# Association between parents' country of birth and multicultural adolescents' psychological well-being in South Korea: A study on depression, worries, life satisfaction, and social withdrawal

**Abdullatif Ghafary**[1]*, **Jaeyong Shin**[2], **Sang Sook Beck**[3], **Jieun Jang**[4], **Rajaguru Vasuki**[5], **So Yoon Kim**[1]

**1** Department of Medical Law and Ethics, Yonsei University, Seoul, South Korea, **2** Department of Preventive Medicine, Yonsei University, Seoul, South Korea, **3** Graduate School of Public Health, Yonsei University, Seoul, South Korea, **4** Department of Preventive Medicine, Dongguk University, Seoul, South Korea, **5** Department of Healthcare Management, Yonsei University, Seoul, South Korea

* Latif.ghafary@yonsei.ac.kr

## Abstract

The purpose of this study was to examine the association between parents' country of birth and psychological well-being of multicultural adolescents in Korea, a country with a predominantly homogenous population. This study used data from the 8th wave of the Multicultural Adolescents Panel Study (MAPS) conducted by the National Youth Policy Institute (NYPI). The participants included 1,147 multicultural adolescents (561 males, 586 females, mean age = 16.96 years). Adolescents whose mothers were born abroad, particularly those whose mothers were Chinese or Filipino, exhibited higher odds of experiencing depression (OR=1.13; 95% CI, 0.50–2.56) compared to those with native Korean mothers. Compared with male adolescents, female respondents were more likely to experience depression (OR = 1.28; 95% CI, 0.99–1.66), worries (OR = 1.98; 95% CI, 1.51–2.59), and lower life satisfaction (OR = 0.75; 95% CI, 0.56–1.01). There was an association between mothers' level of education and adolescents' depression, with higher education levels corresponding to lower depression rates (0R=0.31; 95% CI, 0.13–0.72). These findings have important implications for addressing unique psychological needs in the context of multicultural adolescents, integrated with parental and socioeconomic factors. More support and policy measures should be taken to increase psychological well-being in this growing demographic segment.

**Data availability statement:** The data for this study were obtain from the National Youth Policy Institute (NYPI) and are open data. It can be accessed through the NYPI archive at https://www.nypi.re.kr/archive. In addition, the dataset is included as supporting information with this manuscript for direct access. A readme file is provided to describe the dataset structure, variables, and coding information.

**Funding:** The author(s) received no specific funding for this work.

**Competing interests:** The authors have declared that no competing interests exist.

## Introduction

Within the past two decades, the population of multicultural families, which are composed of married immigrants or foreigners with Korean citizenship, has significantly increased [1]. For clarity, throughout this manuscript, 'Korea' refers specifically to South Korea. The number of students from multicultural families also rose by 7.4% in 2020 to approximately 147,000 students, while the overall number of students in Korea decreased. Adolescents from multicultural families currently constitute 2.2% of the total adolescent population in Korea [1]. Korean adolescents are experiencing significant mental health issues, including depression and suicidal behavior. These issues are influenced by gender, age, and family background [2]. Moreover, multicultural adolescents face additional stressors, such as acculturative stress and discrimination, which escalate these issues [3,4]. The social environment, mental health interventions, and exposure to trauma events play vital roles in life satisfaction [5]. Similarly, for multicultural adolescents, their relationships with parents and the support they receive from the community are crucial for their psychological well-being [6]. Parental background is associated with adolescents' psychological well-being, particularly in multicultural context. Research indicated that sociocultural factors, including parental nationality, education, and socioeconomic status, affect adolescent mental health outcomes [7]. A study on multicultural adolescents in Korea found that acculturative stress, self-esteem, family support, and economic status were the key predictors of mental health challenges [7]. These findings align with study where adolescents from immigrant backgrounds often experience higher rates of anxiety and depression as a consequence of cultural adaptation struggles [8]. The COVID-19 pandemic has intensified mental health problems, leading to increased depression and anxiety. The stress of fitting into a new culture has contributed to some adolescents withdrawing socially [4,9]. Acculturation theory suggests that multicultural adolescents must navigate dual cultural identities, and this can lead to stress and social withdrawal [1]. Studies indicated that language barriers, discrimination, and parental expectations contribute to elevated psychological distress in multicultural adolescents [8]. Additionally, research on gender differences in depression among multicultural adolescents in Korea found that female adolescents reported higher depression rates compared to male adolescents, emphasizing the need for gender-based interventions [8]. The quality of the parent-child relationship is directly associated with adolescents' mental health outcomes, such as depression and worries [10]. Geographical and environmental elements, along with the family's socioeconomic status, are linked with adolescents' risky behaviors [10]. Studies have shown that, compared with their urban counterparts, adolescents in rural areas are more likely to engage in risky behaviors, such as substance use (i.e., drugs, alcohol, and tobacco), bringing weapons to school, and having sexual intercourse [11]. These risky behaviors increase family stress and impact parenting, given that in rural areas, access to mental health care services is limited [12]. The increase in multicultural families has brought about

considerable changes in adolescence and has led to an increase in psychological problems among multicultural families. Adolescents undergo physical, emotional, and social changes, leading to impulsive thoughts and unstable psychological issues such as conflict, tension, and stress among middle and late adolescents, especially among multicultural families [8]. Adolescence is a dynamic period in human development. In addition to accompanying physiological changes, it is characterized by significant psychosocial growth [13]. This period is critical for adolescents, as they begin to explore their identities and positions in society while they grow up physically, psychologically, and socially [14]. The challenges faced by multicultural adolescents in school, the effects of the host country culture, and their relationships with parents can directly and indirectly affect their psychological well-being [15]. Studies indicate that children of immigrants tend to exhibit a higher risk of mental health problems, such as depression and suicidal behavior [16]. In 2020, Korea had the highest suicide rate among OECD member countries [17], revealing the need to act regarding the psychological status of adolescents.

Among the critical factors influencing multicultural adolescents in Korea, the presence of acculturation stress could be associated with increased depressive symptoms and decreased self-esteem [18]. Parental factors, country of birth, level of education, and occupation are some of the significant determining factors among adolescents [15,19].

Addressing the psychological well-being of multicultural adolescents is crucial for promoting social sustainability, and the psychological well-being of future generations is rooted in ethnic diversity [20]. Since adolescence is an irregular period in which biological, cognitive, and social changes occur rapidly, adolescents are vulnerable to developing depression and suicidal symptoms [14]. Previous studies have shown that depression in adolescence is likely to transpire into depression during adulthood [21]. Given these challenges, it is important to understand how parental factors specifically, parent's country of birth is associated with the psychological well-being of multicultural adolescents. Multicultural adolescents in Korea include children from international marriages, immigrant adolescents, and foreign children. This study focuses mainly on children from families with international marriages. To examine these associations, the study assesses depression, worries, life satisfaction, and social withdrawal, while also considering parental education, occupation and socioeconomic status.

## Methodology

### Ethics statement

The study data were obtained from the National Youth Policy Institute (NYPI) after submitting the research plan, and all personal identifiers were excluded to ensure confidentiality.

Although formal IRB approval was not required by NYPI for this specific analysis, informed consent was originally obtained from all participants and their guardians at the time of the initial data collection. All procedures adhered to ethical guidelines, including confidentiality and data protection, as outlined in the Declaration of Helsinki.

### Data set and sample

For this cross-sectional study, we used raw data from the 8th wave of the Multicultural Adolescents Panel Study (MAPS), conducted by the National Youth Policy Institute (NYPI) in Korea, the dataset consists of 1,147 multicultural adolescents (see Fig 1). MAPS was initiated to track the developmental processes of multicultural adolescent [22]. Since the dataset was pre-collected through annual surveys, no independent sampling was performed. Instead, we extracted and structured the dataset to align with our research objectives. For additional details, see S1 Data: Full dataset and codebook. Additionally, for better understanding of the dataset, see S1 Text: Readme. Sample selection was based on data indicating that 95% of the multicultural adolescents in Korea has a foreign mother and a Korean father. Since 2011, annual surveys have investigated 4th grade elementary school students (aged 9–10 years) and their mothers. Further details on the MAPS dataset are available at https://www.nypi.re.kr/archive.

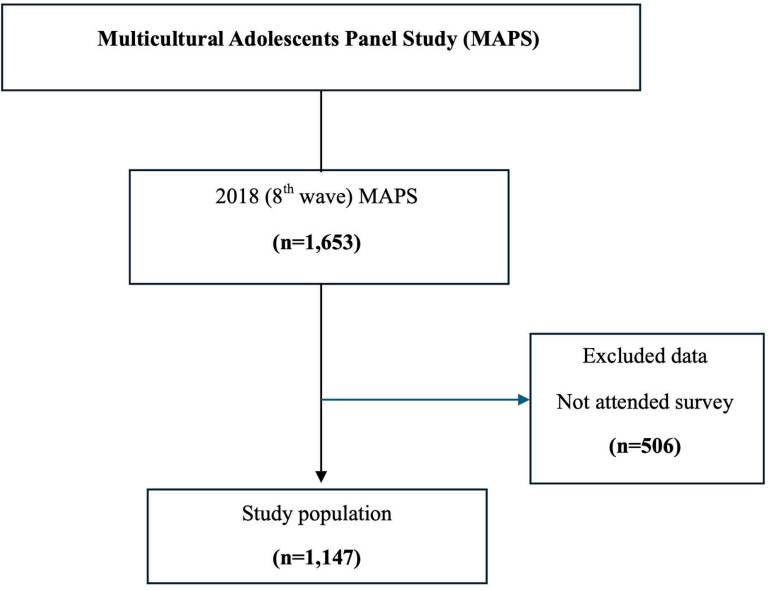

**Fig 1. Flow chart.**

## Data analysis

Data were analyzed using SPSS software version 26.0. Descriptive statistics (percentages, frequencies) categorize the demographics of multicultural adolescents based on gender, age, parental nationality, education, and socioeconomic status. Normality and homogeneity tests were conducted using kurtosis, skewness, and Levene's test.

Multiple logistic regression examined associations between depression, worries, life satisfaction, and social withdrawal in relation to parents' country of birth, age, gender, education level, occupation, marital status, socioeconomic status, and main source of income. Fisher's exact test was used for categorical comparisons assumptions regarding data normality, homogeneity, and validity (see Fig 2).

## Results

### General characteristics

Among the 1,147 participants, 1,109 (96.7%) were foreign mothers, and 1,107 (96.5%) were Korean fathers. The number of participants from the Philippines (25.1%) and China (24.4%) was the highest among foreign mothers. In the current study, 70% of the participants lived in cities, whereas 30% lived in rural areas. A majority (75.2%) answered that fathers were the primary source of income, and 51.5% stated that their families encountered challenging socioeconomic conditions. In addition, 49.7% of the respondents experienced social withdrawal, and 53.7% experienced depression. A total of 36.0% of the respondents were dissatisfied with their future, socioeconomic status, studies, school, work, and physical and psychological well-being. Most parents were high school graduates (see Table 1).

### Associations of depression and worries with the general characteristics of multicultural adolescents

Adolescents whose mothers were Filipino (45.83% depression, 36.8% concern) or Chinese (51.07% depression, 36.1% concern) showed greater depression and concern than those whose mothers were Korean. Higher rates of depression (53.21%) and concern (35.7%) were also found among adolescents whose fathers were Korean.

PLOS Mental Health

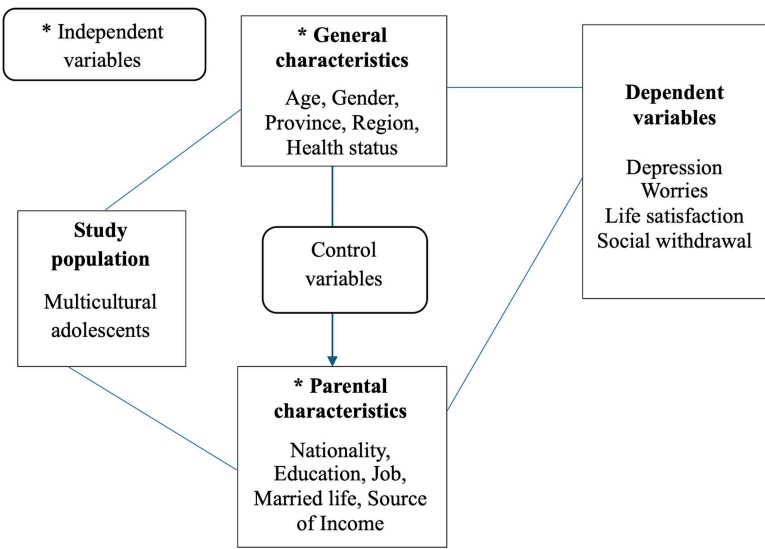

**Fig 2. Conceptual framework.**

Compared with male participants, female participants presented higher rates of depression (57.51%) and concern (44.0%) (see Table 2).

Urban adolescents had a higher rate of depression (55.92%) than did rural adolescents (48.55%), although only 11.5% of urban adolescents worried about their future. Adolescents whose parents had secondary education were more depressed and worried. Higher rates of depression and worries were associated with adolescents whose parents were unemployed, students, or housewives.

### Association of life satisfaction and social withdrawal with the general characteristics of multicultural adolescents

Life satisfaction rates were comparable across different mothers' nationalities. Those with Chinese mothers (77.1% life satisfaction, 52.9% social withdrawal) and Filipino mothers (74.3% life satisfaction, 47.6% social withdrawal) showed greater social withdrawal than did those with Korean mothers (78.9% life satisfaction, 68.4% social withdrawal). However, these differences were not statistically significant (p = 0.834 and p = 0.088, respectively). Compared with non-Korean fathers, adolescents with Korean fathers scored higher in terms of life satisfaction (74.9%) but were more socially withdrawn (49.6%), with a significant difference in social withdrawal (p = 0.008) (see Table 3).

A comparison by gender revealed that boys experienced greater life satisfaction (78.3%, p = 0.007), with no difference in social withdrawal (52.0% boys, 48.6% girls, p = 0.136). Compared with rural adolescents, urban adolescents reported lower life satisfaction (72.5%, p = 0.002) (80.8%), but there was no difference in social withdrawal (50.1% urban, 50.9% rural, p = 0.426).

Parents' education influences both life satisfaction and social withdrawal. Adolescents whose mothers had middle school education or below experienced lower life satisfaction (64.4%, p = 0.02) and greater social withdrawal (37.8%) than did those whose mothers had high school education (76.7%, 47.7%) or undergraduate education or above (72.8%, 54.6%).

Adolescents whose fathers had a high school education reported the lowest levels of life satisfaction (72.2%, p = 0.009) and experienced greater social withdrawal (49.2%). This was in contrast with those whose fathers had middle school or

**Table 1. Characteristics of study participants (N = 1,147).**

| Variables | Categories | N | % |
|---|---|---|---|
| **Mother's country of birth** | Korean | 38 | 3.3 |
| | Non-Korean | 1109 | 96.7 |
| **Mother's nationality** | Korean | 38 | 3.3 |
| | Chinese | 280 | 24.4 |
| | Vietnamese | 25 | 2.2 |
| | Filipino | 288 | 25.1 |
| | Other | 516 | 45.0 |
| **Father's country of birth** | Korean | 1107 | 96.5 |
| | Non-Korean | 40 | 3.5 |
| **Communication with parents** | Korean | 776 | 67.7 |
| | Other language | 371 | 32.3 |
| **Gender** | Male | 561 | 48.9 |
| | Female | 586 | 51.1 |
| **province** | Capital | 397 | 34.6 |
| | Others | 750 | 65.4 |
| **Region** | Urban | 803 | 70.0 |
| | Rural | 344 | 30.0 |
| **Age** | 16 | 84 | 7.3 |
| | 17 | 1023 | 89.2 |
| | 18 | 36 | 3.1 |
| | 19 | 3 | 0.3 |
| | 20 | 1 | 0.1 |
| **Mother's education** | Middle school or less | 45 | 3.9 |
| | High school | 541 | 47.2 |
| | Undergraduate & over | 474 | 41.3 |
| **Father's education** | Middle school or less | 360 | 31.4 |
| | High school | 594 | 51.8 |
| | Undergraduate & over | 193 | 16.8 |
| **Mother's job** | Professional or experts | 358 | 31.2 |
| | Technician or labor | 201 | 17.5 |
| | Others | 588 | 51.3 |
| **Father's job** | Professional or experts | 207 | 18.0 |
| | Technician or labor | 410 | 35.7 |
| | Others | 530 | 46.2 |
| **Parents` married life** | Married | 1059 | 88.5 |
| | Others Ω | 138 | 11.5 |
| **The main source of income** | Father | 862 | 75.2 |
| | Mother | 217 | 18.9 |
| | Others | 68 | 5.9 |
| **Socioeconomic status** | Difficult | 616 | 51.5 |
| | Good | 559 | 46.7 |
| | Missing | 22 | 1.8 |
| **Health status** | Healthy | 1038 | 90.5 |
| | Unhealthy | 109 | 9.5 |
| **Self-esteem** | Not good | 1025 | 89.4 |
| | Good | 122 | 10.6 |

*(Continued)*

PLOS Mental Health

**Table 1.** (Continued)

| Variables | Categories | N | % |
|---|---|---|---|
| **Life satisfaction** | Not good | 287 | 25.0 |
| | Good | 860 | 75.0 |
| **Depression** | Not depressed | 531 | 46.3 |
| | Depressed | 616 | 53.7 |
| **Social withdrawal** | No | 577 | 50.3 |
| | Yes | 570 | 49.7 |
| **Worries and concerns** | No | 734 | 64.0 |
| | Yes | 413 | 36.0 |
| **Academic achievement** | Below average | 399 | 34.8 |
| | Above average | 748 | 65.2 |

less education (79.4% life satisfaction, 46.9% social withdrawal) and those whose fathers had undergraduate or higher education (78.2% life satisfaction, 60.1% social withdrawal, p<0.001). Being in good physical and psychological well-being was a good factor for higher life satisfaction (76.2%, p=0.003) and lower social withdrawal (52.2%, p<0.001) than being in poor health (63.3% life satisfaction, 32.1% social withdrawal).

Academic performance also played a role, with above-average achievers showing greater life satisfaction (80.7%, p<0.001) and lower social withdrawal (56.8%) than below-average performers did (64.2% life satisfaction, 38.1% social withdrawal).

## Multiple logistic regression analysis of factors associated with multicultural adolescents' psychological wellbeing

Adolescents with non-Korean fathers had greater odds of worries (OR=11.42; 95% CI, 1.24–105.35; p=0.032). Female adolescents exhibited higher odds of depression (OR=1.98; 95% CI, 0.99–1.66; p=0.040) and worries (OR=1.98; 95% CI, 1.51–2.59; p<0.001) and lower odds of life satisfaction (OR=0.75; 95% CI, 0.56–1.01; p=0.057) (see Table 4).

Higher mothers' education was linked to lower odds of depression (middle school: OR=0.31; 95% CI, 0.13–0.72, p=0.007; high school: OR=0.32; 95% CI, 0.14–0.77, p=0.010), worries (high school: OR=0.47; 95% CI, 0.23–0.97, p=0.040), and social withdrawal (high school: OR=0.46; 95% CI, 0.22–0.93, p=0.030).

Fathers' education level influenced life satisfaction (middle school: OR=0.59; 95% CI, 0.41–0.85; p=0.004) and social withdrawal (high school: OR=0.62; 95% CI, 0.39–0.97; p=0.035).

Children of labor workers had lower odds of life satisfaction (OR=0.56; 95% CI, 0.36–0.88; p=0.011). Students who derived income from mothers had greater odds of depression (OR=1.79; 95% CI, 1.00–3.19; p=0.039) and worries (OR=1.43; 95% CI, 1.01–2.04; p=0.044).

Using non-Korean languages at home increased the odds of worries (OR=1.63; 95% CI, 1.21–2.18; p<0.001). Unhealthy students had greater odds of depression (OR=2.12; 95% CI, 1.32–3.40; p=0.002), worries (OR=3.40; 95% CI, 2.19–5.27; P<0.001), social withdrawal (OR=2.07; 95% CI, 1.33–3.23; p<0.001), and lower life satisfaction (OR=0.54; 95% CI, 0.35–0.86; p=0.008).

Below-average academic achievement was linked to lower odds of depression (OR=0.55; 95% CI, 0.42–0.73; p<0.001), worries (OR=0.66; 95% CI, 0.50–0.87; p=0.004), and social withdrawal (OR=0.48; 95% CI, 0.36–0.62; p<0.001) but higher life satisfaction (OR=2.62; 95% CI, 1.94–3.54; p<0.001).

**Table 2. Association of depression and worries with general characteristics of multicultural adolescents (N = 1,147).**

| Variables | Categories | Depression | | | | | Worries | | | | |
|---|---|---|---|---|---|---|---|---|---|---|---|
| | | Yes | | No | | p* | Yes | | No | | p* |
| | | N | % | N | % | | N | % | N | % | |
| Mother's nationality | Korean | 25 | 65.79 | 13 | 34.21 | 0.003 | 8 | 32.0 | 17 | 68.0 | 0.972 |
| | Chinese | 143 | 51.07 | 137 | 48.93 | | 101 | 36.1 | 179 | 63.9 | |
| | Vietnamese | 12 | 48.00 | 13 | 52.00 | | 15 | 39.5 | 23 | 60.5 | |
| | Filipino | 132 | 45.83 | 156 | 54.17 | | 106 | 36.8 | 182 | 63.2 | |
| | Other | 304 | 58.91 | 212 | 41.09 | | 183 | 35.5 | 333 | 64.5 | |
| Father's country of birth | Korean | 589 | 53.21 | 518 | 46.79 | 0.051 | 395 | 35.7 | 712 | 64.3 | 0.15 |
| | Non-Korean | 27 | 67.50 | 13 | 32.50 | | 18 | 45.0 | 22 | 55.0 | |
| Gender | Male | 279 | 49.73 | 282 | 50.27 | 0.005 | 155 | 27.6 | 406 | 72.4 | <.001 |
| | Female | 337 | 57.51 | 249 | 42.49 | | 258 | 44.0 | 328 | 56.0 | |
| Region | Urban | 449 | 55.92 | 354 | 44.08 | 0.013 | 92 | 11.5 | 711 | 88.5 | 0.100 |
| | Rural | 167 | 48.55 | 177 | 51.45 | | 30 | 8.7 | 314 | 91.3 | |
| Age | 16 | 46 | 54.76 | 38 | 45.24 | 0.249 | 32 | 38.1 | 52 | 61.9 | 0.052 |
| | 17 | 544 | 53.18 | 479 | 46.82 | | 360 | 35.2 | 663 | 64.8 | |
| | 18+ | 26 | 69.44 | 14 | 30.56 | | 21 | 50.0 | 19 | 50.0 | |
| Province | Capital | 269 | 67.76 | 128 | 32.24 | <.001 | 153 | 38.5 | 244 | 61.5 | 0.109 |
| | Other | 347 | 46.27 | 403 | 53.73 | | 260 | 34.7 | 490 | 65.3 | |
| Mother's education | Middle school or less | 37 | 82.22 | 8 | 17.78 | <.001 | 22 | 48.9 | 23 | 51.1 | 0.102 |
| | High school | 281 | 51.94 | 260 | 48.06 | | 195 | 36.0 | 346 | 64.0 | |
| | Undergraduate & over | 259 | 54.64 | 215 | 45.36 | | 158 | 33.3 | 316 | 66.7 | |
| Father's education | Middle school or less | 175 | 48.61 | 185 | 51.39 | 0.058 | 124 | 34.4 | 236 | 65.6 | 0.606 |
| | High school | 330 | 55.56 | 264 | 44.44 | | 222 | 37.4 | 372 | 62.6 | |
| | Undergraduate & over | 111 | 57.51 | 82 | 42.49 | | 67 | 34.7 | 126 | 65.3 | |
| Mather's job | Professional or experts | 205 | 57.26 | 153 | 42.74 | 0.106 | 121 | 33.8 | 237 | 66.2 | 0.573 |
| | Technician or labor | 113 | 56.22 | 88 | 43.78 | | 75 | 37.3 | 126 | 62.7 | |
| | Others | 298 | 50.68 | 290 | 49.32 | | 217 | 36.9 | 371 | 63.1 | |
| Father's job | Professional or experts | 110 | 53.14 | 97 | 46.86 | 0.838 | 70 | 33.8 | 137 | 66.2 | 0.143 |
| | Technician or labor | 224 | 54.77 | 185 | 45.23 | | 163 | 39.8 | 247 | 60.2 | |
| | Others | 281 | 53.02 | 249 | 46.98 | | 180 | 34.0 | 350 | 66.0 | |
| The main source of income | Father | 446 | 51.74 | 416 | 48.26 | 0.043 | 320 | 37.1 | 542 | 62.9 | 0.369 |
| | Mother | 126 | 58.06 | 91 | 41.94 | | 72 | 33.2 | 145 | 66.8 | |
| | Others | 44 | 64.71 | 24 | 35.29 | | 21 | 30.9 | 47 | 69.1 | |
| Communication with parents | Korean | 401 | 51.68 | 375 | 48.32 | 0.027 | 256 | 33.0 | 520 | 67.0 | <.001 |
| | Other | 215 | 57.95 | 156 | 42.05 | | 157 | 42.3 | 214 | 57.7 | |
| Socioeconomic status | Difficult | 340 | 56.57 | 261 | 43.43 | 0.024 | 224 | 37.3 | 377 | 62.7 | 0.191 |
| | Good | 276 | 50.55 | 270 | 49.45 | | 189 | 34.6 | 357 | 65.4 | |
| Health status | Healthy | 538 | 51.83 | 500 | 48.17 | <.001 | 346 | 33.3 | 692 | 66.7 | <.001 |
| | Unhealthy | 78 | 71.56 | 31 | 28.44 | | 67 | 61.5 | 42 | 38.5 | |
| Academic achievement | Below average | 251 | 62.91 | 148 | 37.09 | <.001 | 166 | 41.6 | 233 | 58.4 | 0.002 |
| | Above average | 365 | 48.80 | 383 | 51.20 | | 247 | 33.0 | 501 | 67.0 | |

**Table 3. Association of life satisfaction and social withdrawal with general characteristics of multicultural adolescents (N = 1,147).**

| Variables | Categories | Life Satisfaction | | | | | Social Withdrawal | | | | |
|---|---|---|---|---|---|---|---|---|---|---|---|
| | | Yes | | No | | p* | Yes | | No | | p* |
| | | N | % | N | % | | N | % | N | % | |
| **Mother's nationality** | Korean | 30 | 78.9 | 8 | 21.1 | | 26 | 68.4 | 12 | 31.6 | 0.088 |
| | Chinese | 216 | 77.1 | 64 | 22.9 | | 148 | 52.9 | 132 | 47.1 | |
| | Vietnamese | 19 | 76.0 | 6 | 24.0 | 0.834 | 15 | 60.0 | 10 | 40.0 | |
| | Filipino | 214 | 74.3 | 74 | 25.7 | | 137 | 47.6 | 151 | 52.4 | |
| | other | 381 | 73.8 | 135 | 26.2 | | 251 | 48.6 | 265 | 51.4 | |
| **Father's country of birth** | Korean | 829 | 74.9 | 278 | 25.1 | 0.436 | 549 | 49.6 | 558 | 50.4 | 0.008 |
| | Non-Korean | 31 | 77.5 | 9 | 22.5 | | 28 | 70.0 | 12 | 30.0 | |
| **Gender** | Boys | 439 | 78.3 | 122 | 21.7 | 0.007 | 292 | 52.0 | 269 | 48.0 | 0.136 |
| | Girls | 421 | 71.8 | 165 | 28.2 | | 285 | 48.6 | 301 | 51.4 | |
| **Region** | Urban | 582 | 72.5 | 221 | 27.5 | 0.002 | 402 | 50.1 | 401 | 49.9 | 0.426 |
| | Rural | 278 | 80.8 | 66 | 19.2 | | 175 | 50.9 | 169 | 49.1 | |
| **Age** | 16 | 60 | 71.4 | 24 | 28.6 | 0.46 | 41 | 48.8 | 43 | 51.2 | 0.402 |
| | 17 | 774 | 75.7 | 249 | 24.3 | | 514 | 50.2 | 509 | 49.8 | |
| | 18 + | 26 | 65.0 | 14 | 35.0 | | 22 | 55.0 | 18 | 45.0 | |
| **Province** | Capital | 262 | 66.0 | 135 | 34.0 | <.001 | 197 | 49.6 | 200 | 50.4 | 0.392 |
| | Other | 598 | 79.7 | 152 | 20.3 | | 380 | 50.7 | 370 | 49.3 | |
| **Mother's education** | Middle school or less | 29 | 64.4 | 16 | 35.6 | | 17 | 37.8 | 28 | 62.2 | |
| | High school | 415 | 76.7 | 126 | 23.3 | 0.105 | 258 | 47.7 | 283 | 52.3 | 0.02 |
| | Undergraduate & over | 345 | 72.8 | 129 | 27.2 | | 259 | 54.6 | 215 | 45.4 | |
| **Father's education** | Middle school or less | 286 | 79.4 | 74 | 20.6 | | 169 | 46.9 | 191 | 53.1 | |
| | High school | 423 | 71.2 | 171 | 28.8 | 0.009 | 292 | 49.2 | 302 | 50.8 | <.001 |
| | Undergraduate & over | 151 | 78.2 | 42 | 21.8 | | 116 | 60.1 | 77 | 39.9 | |
| **Mather's job** | Professional or experts | 276 | 77.1 | 82 | 22.9 | | 185 | 51.7 | 173 | 48.3 | |
| | Technician or labor | 147 | 73.1 | 54 | 26.9 | 0.508 | 89 | 44.3 | 112 | 55.7 | 0.17 |
| | Others | 437 | 74.3 | 151 | 25.7 | | 303 | 54.0 | 258 | 46.0 | |
| **Father's job** | Professional or experts | 167 | 80.7 | 40 | 19.3 | | 111 | 53.6 | 96 | 46.4 | |
| | Technician or labor | 317 | 77.3 | 93 | 22.7 | 0.009 | 190 | 46.3 | 220 | 53.7 | 0.125 |
| | Others | 376 | 70.9 | 154 | 29.1 | | 276 | 52.1 | 254 | 47.9 | |
| **The main source of income** | Father | 646 | 74.9 | 216 | 25.1 | 0.803 | 444 | 51.5 | 418 | 48.5 | 0.359 |
| | Mother | 165 | 76.0 | 52 | 24.0 | | 102 | 47.0 | 115 | 53.0 | |
| | Others | 49 | 72.1 | 19 | 27.9 | | 31 | 45.6 | 37 | 54.4 | |
| **Communication with parents** | Korean | 592 | 76.3 | 184 | 23.7 | 0.080 | 399 | 51.4 | 377 | 48.6 | 0.152 |
| | Other | 268 | 72.2 | 103 | 27.8 | | 178 | 48.0 | 193 | 52.0 | |
| **Socioeconomic status** | Difficult | 447 | 74.4 | 154 | 25.6 | 0.335 | 308 | 51.2 | 293 | 48.8 | 0.271 |
| | Good | 413 | 75.6 | 133 | 24.4 | | 269 | 49.3 | 277 | 50.7 | |
| **Health status** | Healthy | 791 | 76.2 | 247 | 23.8 | 0.003 | 542 | 52.2 | 496 | 47.8 | <.001 |
| | Unhealthy | 69 | 63.3 | 40 | 36.7 | | 35 | 32.1 | 74 | 67.9 | |
| **Academic achievement** | Below average | 256 | 64.2 | 143 | 35.8 | <.001 | 152 | 38.1 | 247 | 61.9 | <.001 |
| | Above average | 604 | 80.7 | 144 | 19.3 | | 425 | 56.8 | 323 | 43.2 | |

**Table 4. Multiple logistic regression analysis of associated factors and multicultural adolescents' psychological well-being (N=1,147).**

| Variables | Categories | Depression OR | 95% CI | p | Worries OR | 95% CI | p | Life Satisfaction OR | 95% CI | p | Social Withdrawal OR | 95% CI | p |
|---|---|---|---|---|---|---|---|---|---|---|---|---|---|
| **Mother's nationality** | Filipino | 1.00 | | | 1.00 | | | 1.00 | | | 1.00 | | |
| | Chinese | 1.13 | 0.50, 2.56 | 0.769 | 1.20 | 0.50, 2.88 | 0.684 | 1.07 | 0.41, 2.78 | 0.896 | 1.34 | 0.58, 3.08 | 0.494 |
| | Korean | 2.08 | 0.74, 5.85 | 0.163 | 1.39 | 0.48, 4.01 | 0.547 | 1.18 | 0.36, 3.95 | 0.783 | 0.69 | 0.24, 1.98 | 0.494 |
| | Vietnamese | 0.92 | 0.40, 2.08 | 0.835 | 1.24 | 0.52, 2.97 | 0.632 | 0.91 | 0.35, 2.37 | 0.852 | 1.65 | 0.72, 3.80 | 0.237 |
| | Other | 1.55 | 0.70, 3.47 | 0.283 | 1.17 | 0.49, 2.76 | 0.724 | 0.89 | 0.35, 2.28 | 0.810 | 1.58 | 0.70, 3.59 | 0.271 |
| **Father's country of birth** | Korean | 1.00 | | | 1.00 | | | 1.00 | | | 1.00 | | |
| | Non-Korean | 1.88 | 0.26, 13.47 | 0.530 | 11.42 | 1.24, 105.35 | 0.032 | 0.78 | 0.13, 4.74 | 0.785 | 0.44 | 0.08, 2.58 | 0.363 |
| **Gender** | Male | 1.00 | | | 1.00 | | | 1.00 | | | 1.00 | | |
| | Female | 1.28 | 0.99, 1.66 | 0.040 | 1.98 | 1.51, 2.59 | <.001 | 0.75 | 0.56, 1.01 | 0.057 | 1.11 | 0.86, 1.43 | 0.408 |
| **Region** | Urban | 1.00 | | | 1.00 | | | 1.00 | | | 1.00 | | |
| | Rural | 0.97 | 0.71, 1.32 | 0.843 | 1.06 | 0.77, 1.46 | 0.730 | 1.20 | 0.82, 1.74 | 0.350 | 0.94 | 0.69, 1.27 | 0.679 |
| **Mother's education** | Undergraduate & over | 1.00 | | | 1.00 | | | 1.00 | | | 1.00 | | |
| | Middle school or less | 0.31 | 0.13, 0.72 | 0.007 | 0.56 | 0.28, 1.13 | 0.104 | 1.20 | 0.59, 2.45 | 0.616 | 0.63 | 0.32, 1.24 | 0.153 |
| | High school | 0.32 | 0.14, 0.77 | 0.010 | 0.47 | 0.23, 0.97 | 0.040 | 0.89 | 0.42, 1.88 | 0.762 | 0.46 | 0.22, 0.93 | 0.030 |
| **Father's education** | Undergraduate & over | 1.00 | | | 1.00 | | | 1.00 | | | 1.00 | | |
| | Middle school or less | 1.12 | 0.82, 1.52 | 0.468 | 1.18 | 0.85, 1.62 | 0.324 | 0.59 | 0.41, 0.85 | 0.004 | 0.86 | 0.64, 1.16 | 0.332 |
| | High school | 1.16 | 0.74, 1.84 | 0.516 | 1.21 | 0.75, 1.94 | 0.441 | 0.81 | 0.48, 1.39 | 0.448 | 0.62 | 0.39, 0.97 | 0.035 |
| **Mother's job** | Professional or experts | 1.00 | | | 1.00 | | | 1.00 | | | 1.00 | | |
| | Technician or labor | 0.80 | 0.59, 1.08 | 0.148 | 1.20 | 0.88, 1.65 | 0.243 | 0.74 | 0.53, 1.05 | 0.093 | 0.91 | 0.68, 1.22 | 0.527 |
| | Others | 1.14 | 0.75, 1.72 | 0.550 | 1.02 | 0.67, 1.57 | 0.923 | 0.62 | 0.39, 0.99 | 0.053 | 1.23 | 0.82, 1.85 | 0.308 |
| **Father's job** | Professional or experts | 1.00 | | | 1.00 | | | 1.00 | | | 1.00 | | |
| | Technician or labor | 1.11 | 0.76, 1.64 | 0.588 | 1.01 | 0.68, 1.51 | 0.955 | 0.56 | 0.36, 0.88 | 0.011 | 0.76 | 0.52, 1.11 | 0.162 |
| | Others | 1.40 | 0.92, 2.14 | 0.114 | 1.37 | 0.89, 2.10 | 0.154 | 0.75 | 0.46, 1.23 | 0.248 | 0.96 | 0.64, 1.44 | 0.840 |
| **The main source of income** | Father | 1.00 | | | 1.00 | | | 1.00 | | | 1.00 | | |
| | Mother | 1.79 | 1.00, 3.19 | 0.039 | 0.77 | 0.42, 1.39 | 0.380 | 1.10 | 0.60, 2.02 | 0.762 | 1.55 | 0.90, 2.69 | 0.115 |
| | Others | 1.43 | 1.01, 2.04 | 0.044 | 0.77 | 0.53, 1.10 | 0.153 | 0.98 | 0.66, 1.45 | 0.908 | 1.21 | 0.86, 1.70 | 0.272 |
| **Communication with parents** | Korean | 1.00 | | | 1.00 | | | 1.00 | | | 1.00 | | |
| | Other | 1.13 | 0.84, 1.50 | 0.422 | 1.63 | 1.21, 2.18 | <.001 | 0.86 | 0.62, 1.20 | 0.378 | 1.27 | 0.95, 1.68 | 0.102 |
| **Socioeconomic status** | Difficult | 0.76 | 0.59, 0.99 | 0.042 | 0.79 | 0.61, 1.04 | 0.093 | 1.24 | 0.92, 1.66 | 0.157 | 1.06 | 0.82, 1.36 | 0.670 |
| | Good | 1.00 | | | 1.00 | | | 1.00 | | | 1.00 | | |
| **Health status** | Healthy | 1.00 | | | 1.00 | | | 1.00 | | | 1.00 | | |
| | Unhealthy | 2.12 | 1.32, 3.40 | 0.002 | 3.40 | 2.19, 5.27 | <.001 | 0.54 | 0.35, 0.86 | 0.008 | 2.07 | 1.33, 3.23 | <.001 |
| **Academic achievement** | Below average | 0.55 | 0.42, 0.73 | <.001 | 0.66 | 0.50, 0.87 | 0.004 | 2.62 | 1.94, 3.54 | <.001 | 0.48 | 0.36, 0.62 | <.001 |
| | Above average | 1.00 | | | 1.00 | | | 1.00 | | | 1.00 | | |

## Discussion

This study investigated how parental factors specifically, parent's country of birth is associated with the psychological well-being of multicultural adolescents in Korea, with a specific focus on depression, worries, life satisfaction, and social withdrawal. The sample included 1,147 adolescents, with the majority having foreign-born mothers and Korean fathers. The findings indicated that adolescents whose mothers were Chinese or Filipino were more likely to experience greater depression, worries, and social withdrawal. While mothers' nationalities made minimal association in life satisfaction rates, adolescents of Korean mothers reported marginally greater life satisfaction rate compared to their Filipino and Chinese counterparts. Given the small sample size of Korean mothers (N = 38), this comparison should be viewed cautiously. Similarly, having a foreign-born father was linked to increased worries. This could be explained by several factors. Adolescents of immigrants experience many conflicts and unfamiliar situations as they grow up in a dual culture where the values and attitudes of their fathers and mothers are different. Gender differences revealed that females had greater odds of depression, worries and lower life satisfaction. Higher parental education, particularly among mothers, was associated with lower odds of depression and worries. Fathers' occupations had a noticeable association on life satisfaction, with adolescent of labor workers reporting lower life satisfaction. Additionally, adolescents with mothers from developing countries tended to have greater depressive symptoms. Many immigrants from these countries are often employed in low-paying, difficult, and undesirable jobs that Koreans refuse, which may be associated with lower socioeconomic status. Language used to communicate with parents is associated with language competency. In this study, we found that adolescents who spoke non-Korean languages at home were more likely to have worries and concerned about their lives. Additionally, students who had poor health had a greater risk of developing depression, worries, and social withdrawal and lower life satisfaction. Surprisingly, those who had poorer academic achievements were less likely to suffer from depression and worries and had greater life satisfaction. This paradoxical finding may be linked to the pressures and expectations associated with academic success in Korea.

These findings have important implications for addressing unique psychological needs in the context of multicultural adolescents, integrated with parental and socioeconomic factors. More support and policy measures should be taken to increase psychological well-being in this growing demographic segment.

## Limitations

Despite the strengths of this study, several limitations should be acknowledged. While the data were collected through home visits by trained interviewers, they were based on self-reports from the participants and may have response bias. Additionally, the study was based on a cross-sectional survey, so causality could not be confirmed; only associations were identified. Finally, while the results indicate that the country of birth, job, education, and socioeconomic status of parents are associated with the psychological well-being of multicultural adolescents, these associations are not statistically significant.

## Conclusion

This study provides valuable insights into the association of parents' country of birth and multicultural adolescents psychological well-being in Korea. Key findings revealed that adolescents whose mothers were foreign-born, especially those from China and the Philippines, were more likely to experience higher levels of depression, worries, and social withdrawal and lower life satisfaction. Consequential effects on the psychological well-being of adolescents include the father's occupation and the use of languages other than Korean at home. These findings suggest that interventions may be beneficial for assisting multicultural adolescents and addressing language barriers, socioeconomic hurdles, and cultural integration.

With an in-depth understanding of these specific factors, it becomes easier to formulate practical approaches that positively impact the psychological well-being of multicultural adolescents.

## Supporting information

**S1 Data. This file contains the full dataset used in the study, including anonymized responses from participants. Variables are coded according to the study methodology.**
(XLSX)

**S1 Text. Readme provides an overview of the supporting information files in this study. It explains the contents of the dataset and the structure of different sheets within the associated Excel file, S1 Data.**
(PDF)

## Acknowledgments

We would like to express our deepest gratitude to all those who supported and contributed to this research. Special thanks to the National Youth Policy Institute (NYPI) for providing the data essential to this study.

## Author contributions

**Conceptualization:** Abdullatif Ghafary, Jaeyong Shin.

**Formal analysis:** Abdullatif Ghafary, Rajaguru Vasuki.

**Investigation:** Jaeyong Shin, Sang Sook Beck.

**Methodology:** Abdullatif Ghafary, Jaeyong Shin.

**Project administration:** Jieun Jang.

**Software:** Abdullatif Ghafary, Jieun Jang.

**Supervision:** Jaeyong Shin, Sang Sook Beck, Jieun Jang, Rajaguru Vasuki, So Yoon Kim.

**Writing – original draft:** Abdullatif Ghafary.

**Writing – review & editing:** Abdullatif Ghafary, Rajaguru Vasuki.

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
