## [Decision Letter · Decision Letter 0]

PMEN-D-25-00105

Impact of parents' country of birth on multicultural adolescents' wellbeing in South Korea: A study on depression, worries, life Satisfaction, and social Withdrawal

PLOS Mental Health

Dear Dr. Ghafary,

Thank you for submitting your manuscript to PLOS Mental Health. After careful consideration, we feel that it has merit but does not fully meet PLOS Mental Health’s publication criteria as it currently stands. Therefore, we invite you to submit a revised version of the manuscript that addresses the points raised during the review process.

We look forward to receiving your revised manuscript.

Kind regards,

Hanif Abdul Rahman, Ph.D.

Academic Editor

PLOS Mental Health

Additional Editor Comments (if provided):

Reviewers' comments:

Reviewer's Responses to Questions

**Comments to the Author**

1. Does this manuscript meet PLOS Mental Health’s publication criteria?

Reviewer #1: Yes

Reviewer #2: No

2. Has the statistical analysis been performed appropriately and rigorously?

Reviewer #1: Yes

Reviewer #2: No

3. Have the authors made all data underlying the findings in their manuscript fully available (please refer to the Data Availability Statement at the start of the manuscript PDF file)?

Reviewer #1: Yes

Reviewer #2: No

4. Is the manuscript presented in an intelligible fashion and written in standard English?

Reviewer #1: Yes

Reviewer #2: Yes

Reviewer #1: Great job on this paper! Thank you for this amazing work. The study design is very rigorous and allows for great causality and analysis. Here are some areas for edits:

- There are some consistency issues in paragraph formatting and in the title with different capitalization

- I think your study focus would be more effective at the end of your introduction

- Figure 1 became a bit oddly formatted and stretched so it will need to be re-edited into the manuscript

- Figure 2 also formatted oddly

- Line 172: “Adolescents with foreign mothers generally reported higher life satisfaction” versus 228-230: “The findings revealed that adolescents with mothers from the Philippines and China had higher levels of depression, worry, and social withdrawal, as well as lower life satisfaction, than did those with native Korean mothers” These statements seem to be contradicting

Reviewer #2: The research is simply a fishing expedition. The analysis reflects patterns that the authors infer to be causal despite a purely non-experimental research design. There is no literature review to substantiate a clear purpose. A clear research question was not articulated, let alone methodology. The quality of writing was fine, I just don't think this article belongs in a journal, even with major revisions.

**Do you want your identity to be public for this peer review?** For information about this choice, including consent withdrawal, please see our Privacy Policy

Reviewer #1: No

Reviewer #2: No

---

## [Editor Report · Decision Letter 1]

Association between parents' country of birth and multicultural adolescents' psychological well-being in South Korea: A study on depression, worries, life satisfaction, and social withdrawal

PMEN-D-25-00105R1

Dear Mr. Ghafary,

We are pleased to inform you that your manuscript 'Association between parents' country of birth and multicultural adolescents' psychological well-being in South Korea: A study on depression, worries, life satisfaction, and social withdrawal' has been provisionally accepted for publication in PLOS Mental Health.

Best regards,

Hanif Abdul Rahman, Ph.D.

Academic Editor

PLOS Mental Health